# Combination Therapy for Neuropathic Pain: A Review of Recent Evidence

**DOI:** 10.3390/jcm10163533

**Published:** 2021-08-11

**Authors:** Ancor Serrano Afonso, Thiago Carnaval, Sebastià Videla Cés

**Affiliations:** 1Department of Anesthesiology, Resuscitation and Pain Management, Hospital Universitari de Bellvitge, 08907 L’Hospitalet de Llobregat, Spain; 2Department of Clinical Pharmacology, Hospital Universitari de Bellvitge, 08907 L’Hospitalet de Llobregat, Spain; thiagocarnaval@gmail.com; 3Pharmacology Unit, Department of Pathology and Experimental Therapeutics, School of Medicine and Health Sciences, IDIBELL, University of Barcelona, 08907 L’Hospitalet de Llobregat, Spain; svidela@ub.edu

**Keywords:** neuropathic pain, combination therapy, pharmacotherapy, randomized control trial

## Abstract

Pharmacological treatment is not very effective for neuropathic pain (NP). A progressive decrease in the estimated effect of NP drugs has been reported, giving rise to an increase in the use of the multimodal analgesic approach. We performed a new independent review to assess whether more and better-quality evidence has become available since the last systematic review. We evaluated the efficacy, tolerability, and safety of double-blinded randomized controlled trials involving only adult participants and comparing combination therapy (CT: ≥2 drugs) with a placebo and/or at least one other comparator with an NP indication. The primary outcome assessed was the proportion of participants reporting ≥50% pain reductions from baseline. The secondary outcome assessed was the proportion of drop-outs due to treatment-emergent adverse events. After removing duplicates, 2323 citations were screened, with 164 articles assessed for eligibility, from which 16 were included for qualitative analysis. From the latter, only five lasted for at least 12 weeks and only six complied with the required data for complete analysis. CT has been adopted for years without robust evidence. Efforts have been made to achieve better-quality evidence, but the quality has not improved over the years. In this regard, guidelines for NP should attempt to make recommendations about CT research, prioritizing which combinations to analyze.

## 1. Introduction

Neuropathic pain (NP) occurs as a direct consequence of an injury or disease that affects the somatosensory system [1]. The prevalence of NP in the population varies from 6.9% to 10%, depending on the tool used for its diagnosis [2], and it negatively affects quality of life, impacting daily activities, such as sleeping and walking, and family and social interactions. [3]. Patients with uncontrolled pain continuously suffer heavy individual and societal burdens, which could make them believe that chronic pain is inevitable and untreatable, especially for those who are not responding to standard measures. A considerable number of patients do not achieve a satisfactory pain relief or improvements in their quality of life with currently available drugs [4]. Pharmacological therapy remains an important component of NP management [5,6]. However, more than a decade has passed since the market release of the last drug suggested for NP treatment, according to international guidelines. Clinical guidelines recommend starting treatment with monotherapy and placing combination treatment (CT) in a second tier for patients who do not respond to monotherapy or switching [7,8].

The treatment of NP is effective in less than 50% of patients and is also associated with significant adverse drug effects [9]. In addition, decreases in drug effect have been reported across all drug classes, with a progressive increase in the number needed to treat (NNT) in randomized control trials (RCTs) [10]. The reason for this increase in NNT numbers is not well known. This is probably due to a combination of different causes: more complex trial designs required by regulatory agencies, such as the US Food and Drug Administration (FDA) or the European Medicines Agency (EMA), with larger sample sizes, longer study periods, better randomization and blinding reports, and intention-to-treat (ITT) analysis [11,12,13]; more elaborated efficacy reports with an increased goal (i.e., the use of 30% to 50% pain reduction as outcome measures) [10,14]; and the contribution of other factors for higher levels of placebo response in NP RCTs [15].

Therefore, CT is becoming more and more popular among clinicians [16,17,18,19], and its rationale lies on two theories: (1) a phenotypic profile-guided treatment improves symptomatic control (i.e., different clinical signs and symptoms are suggested to reflect different underlying mechanisms) and offers the possibility of an individualized mechanism-based treatment approach for different somatosensory patterns [20,21,22]; (2) targeting more than one NP mechanism simultaneously with CT could be a better approach than targeting a single mechanism with a single drug [23,24] since it may allow lower doses of individual drugs (due to a synergistic effect) and improve their safety/tolerability profile.

However, doubts continue to arise regarding which drug combination is effective or which drugs to combine. For instance, some guidelines recommend adding an agent from another class if pain control is inadequate [25]. Others state that the insufficient data do not allow them to make any CT recommendations [17,26] or that the evidence for combinations is inconclusive [8], with some recent recommendations for certain combinations having weak evidence [8,16]. The last Cochrane review carried out in 2012 already indicated that, somewhat surprisingly, they were able to identify only 107 relevant citations, from which only 21 could be considered high-quality evidence on the matter [27].

Considering the decreasing estimates in drug effect in NP RCTs [10], the 9-year gap since the last published Cochrane review (the previous one was published 7 years earlier, in 2005, by Gilron et al. [28]), and the absence of a consensus among clinicians regarding when to start CT and which drugs to combine, we thought it was appropriate to perform a new and independent systematic review.

Hence, we performed a new independent review, searching medical databases from 2012 onwards (due to the aforementioned increases in NNT values and changes in trials’ designs). We aimed to assess the availability and quality of evidence regarding CT for NP. Here, we make recommendations regarding CT options and propose a treatment algorithm to guide future therapeutic decisions.

## 2. Materials and Methods

Even though this is a new independent review, we followed the previous systematic review methods as closely as possible [27]. We evaluated the efficacy, tolerability, and safety of various drug combinations for the treatment of NP. For that purpose, we identified only RCTs of various drug combinations for NP from different databases. We also hand searched for citations within other reviews and trial registries. The most recent search was performed on 30 April 2021.

### 2.1. Criteria for Study Selection


We applied the following criteria when selecting studies for the qualitative analysis.

### 2.2. Types of Studies


We sought out only double-blind RCTs for the treatment of NP that compared combinations of two or more drugs with a placebo and/or at least one other comparator with NP indication.

### 2.3. Participants


We only included studies involving adult participants 18 years and older with a diagnosis of NP.

### 2.4. Interventions


We included interventions involving a combination of two or more different drugs. We did not include any studies performed with non-pharmacological treatments (even if they were interventional), such as diets (including vitamin supplements) or physical measures.

### 2.5. Outcomes


#### 2.5.1. Primary Outcomes


The primary outcome we assessed was the proportion of participants reporting ≥50% pain reduction from baseline. When 50% pain reduction was not reported, we looked for a decrease in pain by ≥30% from the baseline.

#### 2.5.2. Secondary Outcomes


We looked for (i) the proportion of drop-outs due to treatment-emergent adverse effects and (ii) the proportion of participants reporting each specific adverse effect (i.e., sedation and dizziness) with moderate or greater severity. 

### 2.6. Search Methods


We searched the following databases, timelines, and restrictions:
PubMed^®^. Search keywords: (“neuropathic pain” AND “combination”). The timeline was limited to articles published from 1 January 2012 to 3 March 2021. Language filter: English. The last search was performed on 3 March 2021.Google Scholar. Search keywords: (“neuropathic pain” AND “combination therapy”). The timeline was limited to articles published from 2012 to 2021. Language filter: English. The last search was performed on 15 March 2021.Web of Science. Search keywords: (“neuropathic pain” AND “combination therapy”). The search was performed on all databases except for the zoological one. The timeline was limited to articles published from 2012 to 2021. Filters were applied to exclude review articles, case reports, editorial material, books, meetings, and letters and corrections. Language filter: English. The last search was performed on 30 April 2021.SCOPUS. Search keywords: (“neuropathic pain” AND combination) in the title, abstract, or keywords. The timeline was limited to publication years after 2011. Filters: document type “articles” and only English language. The last search was performed on 30 April 2021.We further searched ClinicalTrials.gov. Keywords: (“neuropathic pain” AND “combination therapy”). No filters were used. The last search was performed on 3 March 2021.We searched within the reference lists of all the included studies.Finally, we also checked for relevant citations within other reviews and meta-analyses published between 2012 and 2021.

### 2.7. Data Collection and Analysis


All citations were screened by title and, when not directly excluded, by abstract to be assessed for eligibility by the corresponding author (A.S.A.). From those assessed for eligibility, A.S.A. and S.V. performed a second thorough selection for qualitative analysis. All selected studies were checked independently by the three authors for criteria fulfillment for qualitative analysis, and a triple cross-check was performed afterwards. Data extraction from selected studies was performed by S.V. and T.C. and, again, cross-checked afterwards.

#### 2.7.1. Data Extraction


From each selected study, using the aforementioned criteria, we extracted the following data:
The proportion of participants with 50% pain relief (primary outcome);The proportion of participants with 30% pain relief (whenever 50% was not reported and even if 50% was reported);The proportion of drop-outs due to treatment-emergent adverse events (secondary outcome);The proportion of dropouts for any other reason (secondary outcome);The proportion of participants reporting each specific adverse effect (i.e., sedation and dizziness) with moderate or greater severity; andThe study drugs, including the names, doses, routes of administration, and treatment durations.

#### 2.7.2. Risk of Bias


We searched for the following types of bias in all of the studies included for qualitative analysis: random sequence generation and allocation concealment (selection bias), blinding, incomplete outcome data, selective reporting, and other potential sources of bias. We graded all selected studies for quality per the Cochrane Risk of Bias tool [29].

#### 2.7.3. Measures of Effect


We sought for comparative effects between CT study drugs and their comparators, which could be either a placebo or the individual drugs that comprise the CT.

#### 2.7.4. Unit of Analysis


For those studies with more than one active treatment group, we divided the control group among the active treatment arms to allow comparison between them.

#### 2.7.5. Missing Data


We analyzed the data based on ITT. We considered all randomized patients in the ITT population who received assigned treatments that provided at least 50% of the required outcome data.

#### 2.7.6. Heterogeneity


To avoid heterogeneity, we did not assess any study that did not have similar conditions for analysis. 

#### 2.7.7. Groups and Subgroups


We looked for any subgroup that could produce a different combination of study results (i.e., phenotyping). Finally, for the discussion, even if grouping was not possible, we categorized studies according to the drug classes used for CT (i.e., opioids, antidepressants, anticonvulsants, etc.).

## 3. Results

### 3.1. Description of Studies


The steps taken during this research are summarized in
Figure 1
. We identified a total of 3808 citations, including records from databases and additional records from other sources, ending the search on 30 April 2021. After removing duplicate citations, we ended up with 2323 individual citations to screen. After thorough screening, we assessed a total of 164 articles for eligibility, out of which 16 were included for qualitative analysis. Only six of them complied with the requirements of data (primary outcome) for complete analysis. No data could be added up or combined for quantitative analysis. Hence, no meta-analysis could be conducted.

#### 3.1.1. Study Selection


We identified 16 studies that fulfilled the inclusion criteria for this review: RCTs, double-blinded, and a comparison of combinations of two or more drugs with a placebo and/or at least one other comparator for the treatment of NP [30,31,32,33,34,35,36,37,38,39,40,41,42,43,44,45]. Among them, only six provided data on the primary outcome (proportion of participants reporting ≥50% or ≥30% pain reductions from baseline), either by direct reporting or by deduction through study figures or graphs (data from such studies can be seen in Table 1). In total, 1243 participants were included in the study drugs groups vs. 928 were included in the control groups: one RCT evaluated the combination of cannabinoids delta-9-tetrahydrocannabinol (THC)/cannabidiol (CBD) oromucosal spray and the existing treatment regimen for central neuropathic pain (CNP) in patients with multiple sclerosis [43]; a different drug combination (opioid plus pregabalin (PGB) plus duloxetine (DXT) was tested in one RCT in NP in cancer patients [32,40]; one tested a combination of DXT and PGB against both of them on monotherapy in painful diabetic neuropathy (PDN) [42]; another compared a drug combination of dextromethorphan and quinidine against a placebo, again in PDN [45]; and capsaicin 8% dermal patch (CP8) in combination with systemic NP medications was evaluated in another RCT in postherpetic neuralgia (PHN) [44]. Likewise, these studies also provided data on the secondary outcomes: (i) the proportion of participants dropping out of the study due to treatment-emergent adverse effects and (ii) the proportion of participants reporting each specific adverse effect (i.e., sedation and dizziness) with moderate or greater severity. 

#### 3.1.2. Study Design


Among the selected RCTs, twelve studies [30,32,33,34,35,36,39,40,42,43,44,45] used a parallel design and four [31,37,38,41] used a crossover design. None of the crossover trials conducted analyses involving only first period data, likely due to inadequate statistical power. 

Among the six RCTs that provided data on the primary outcome, three compared the combination of interest with the placebo alone [43,44,45]; one compared a combination of two drugs against monotherapy of each drug and the placebo [38]; another one compared CT only against high-dose monotherapy of each, with no placebo control [42]; and the last one compared the combination of three painkiller drugs with the combination of only two of these painkiller drugs in cancer patients with NP [32]. It is noteworthy that only three of them had a treatment period of at least 12 weeks, excluding the titration period [43,44,45].

#### 3.1.3. Outcomes


Five studies reported the number of patients with a ≥50% pain reduction [38,42,43,44,45], and in other studies, this number was deduced from the figures [32]. Most of these studies also reported the number of patients with ≥30% pain reduction, except for two studies [38,43]. One study described the proportion of patients reporting ≥50% pain reductions and ≥30% pain reductions, but these proportions were assessed using a secondary analysis producing the overall percentage from all treatment groups on a three-branch crossover study. The number of participants could not be calculated from this percentage or from the diagram of participants included and withdrawn from the study [37]. Other outcomes such as adverse effects and the patient’s overall impression of the change in pain relief are shown in a table in the Appendix A.

According to the guideline on the clinical development of medicinal products intended for the treatment of pain [46], a sustained therapeutic effect in chronic pain should, in general, be demonstrated in pivotal efficacy trials with a treatment period of at least 12 weeks [47]. Five out of sixteen selected RCTs provided data on a period of at least 12 weeks [30,34,43,44,45], all with a parallel design.

#### 3.1.4. Pain Conditions


PDN was explored in three studies [30,42,45], PHN was explored in one study [44], neuropathic cancer pain (N-CP) was explored in three studies [32,33,40], lumbar spinal stenosis (spinal cord injury (SCI) pain) or low back pain were explored in two studies [35,36], CNP was explored in two studies [39,43], other different neuropathic conditions were evaluated in four RCTs [34,37,38,41], and one condition was evaluated in long-standing NP [31].

#### 3.1.5. Excluded Studies


For the purpose of this review, we did not include any other intervention that was not about drug CT for NP. Thus, independent of whether they were RCTs or not, all studies that used other comparators such as diet, vitamins, non-medical therapy (i.e., physical therapy), and any kind of interventional therapy (i.e., neuraxial, nerve blocks, etc.) were not included for analysis. We also excluded studies that compared CT for NP but were not RCTs (i.e., observational analysis, cohort studies, retrospective analysis, open label, etc.). Post hoc analyses of other RCTs were excluded too. 

Of the 16 selected RCTs that fulfilled the inclusion criteria of this review, 10 were excluded because they did not provide data on the primary outcome [30,31,33,34,35,36,37,39,41,45]. Even if an RCT reported other pain assessments (such as Patient Global Impression of Change (PGIC), Brief Pain Inventory (BPI), or both) but not the primary endpoint (≥50% reductions in pain from baseline), when it was not reported, then that RCT did not meet the criteria to be included in the full analysis. The data on these non-selected studies are shown in Table 2. Therefore, this review focused only on six studies [32,40,42,43,44,45], from which only three RCTs provided data from a period of at least 12 weeks [43,44,45].

### 3.2. Risk of Bias

The risk of bias is shown in Table 3. Judgements about each item in the assessment of risk of bias presented as percentages across studies can be found in the Appendix A.

#### 3.2.1. Random Sequence Generation and Allocation Concealment (Selection Bias)


Four of the six studies reported the method used to generate the random sequence and to keep the allocation concealed [32,38,42,44]. The other two appropriately reported only one or the other [43,45].

#### 3.2.2. Blinding

Only one study [40] was not blinded. Among the other studies, although all of them claimed to be blinded, 5 out of 15 studies [30,35,36,41,45] did not describe the blinding procedure.

#### 3.2.3. Incomplete Outcome Data

We qualified attrition bias as “low risk” for studies where the dropout rate was below 20%. We qualified studies with higher dropout rates but included ITT analyses as “unclear” or “high risk of bias”. All six studies provided information about trial dropouts.

#### 3.2.4. Selective Reporting


Although four out of the six selected studies [32,38,42,43] indicated pre-trial registration on a clinical trial registry, all six of them reported on at least one of the outcomes that was considered to be clinically relevant.

#### 3.2.5. Other Potential Sources of Bias


We assessed the issue of other bias as being high risk in studies where the follow-up was shorter than twelve weeks [32,38,42] and/or where the study had fewer than 50 participants per arm or period of treatment in parallel or crossover studies, respectively [32,38].

### 3.3. Effect of Interventions


When combining THC/CBD oromucosal spray as an add-on with a pre-existing regimen for central neuropathic pain, the number of responders (≥50% pain reduction from baseline) at week 10 totaled 30% in the THC/CBD spray group compared with 28% in the placebo group (*p*-value: 0.714). Therefore, there was no difference between treatment groups, mostly due to a similar (high) number of placebo responders [43].

For a 12-week period, in patients who used systemic pain medication, CP8 as an add-on therapy reduced NP in PHN. The number of responders (≥50% pain reduction from baseline) was 29% in the CP8 group and 17% in the placebo group (p-value: 0.048). Likewise, CP8 reduced NP in PHN in patients who did not use systemic medication (36% for CP8 group, 25% for placebo group, *p*-value: 0.004) [44]. We could not differentiate between concomitant medications. No data in this regard were given.

A combination of dextromethorphan and quinidine was effective, with an acceptable safety profile, for the treatment of PDN pain. The proportion of patients who achieved a 50% rating scale score reduction was 66% (DMQ 45/30 mg twice a day) and 54% (DMQ 30/30 mg twice a day), compared with the placebo group (*p*-value: 0.001 and 0.006, respectively) [45].

A combination of PGB+DXT showed no significant differences in pain reduction in patients with PDN when compared with either PGB or DXT at high-dose monotherapy. The number of responders (≥50% pain reduction from baseline) was 52% in the “60 mg DLX + 300 mg PGB” group, 51% in the “300 mg PGB + 60 mg DLX” group, 29% in the “60 mg DLX + 60 mg DLX” group, and 47% in the “300 mg PGB + 300 mg PGB” group (*p*-value: 0.068) [42].

A combination of moderate doses of the tricyclic antidepressant (ATC) imipramine and PGB could be considered an alternative to high-dosage monotherapy for painful polyneuropathy. The percentage of patients who had at least 50% relief with respect to baseline was placebo 6%, PGB 12%, imipramine 20%, and CT 28% [38].

Adding DXT to opioid–PGB therapy might have clinical benefits in alleviating refractory N-CP. The proportion of patients who achieved 30% or more pain reduction was 37.5% in patients eventually receiving PGB and DXT, 60% in those receiving DXT only, 23.1% in those receiving PGB only, and 0.00% (90% CI 0.00–40,96) in those receiving placebo only [32]. However, this effect was assessed only for 10-day therapy. No information was presented for a longer period of treatment.

## 4. Discussion

In this review we tried to identify new RCTs that could bring new evidence on CT for NP published after the last Cochrane’s systematic review in 2012 [27]. After a thorough data search from various databases, we could only find 16 RCTs, of which only 5 directly reported the primary outcome and, in 1 RCT, the primary outcome had to be deduced from the figures. In addition, out of the 16, only 5 RCTs reported pain outcomes after 12 weeks [30,34,43,44,45], as required by the standards. We were not able to perform a quantitative analysis (meta-analysis). The main reasons were the small number of trials included and the heterogeneity among them, not only on the drug types and combinations used, but also on type of pain and type of study, which made accomplishing a meta-analysis impossible.

### 4.1. Main Results


We can assume that adding THC/CBD to a pre-existing treatment for NP has no effect and that no difference between a PGB–DXT combination exists against either drug on high-dose monotherapy. However, a combination of an ATC such as imipramine and PGB may be an alternative to high-dose monotherapy. Likewise, adding DXT to a previous opioid–PGB therapy may be beneficial too, and topical CP8 for peripheral NP is effective in reducing NP regardless of the concomitant therapy. Different unusual combinations such as dextromethorphan and quinidine may be another useful treatment option. 

For the secondary outcomes, all the selected studies had safety reports, where they differentiated adverse effects and drop-outs due to treatment-emergent adverse effects (only one for the latter) [32]. However, the data on prescribed rescue medication were either not available [42] or not analyzed [42,43,44]. Other pain ratings or sleep interferences were evaluated in some studies. A PGIC was evaluated in three studies [42,43,44], and the BPI was evaluated also in three studies [32,42,43]. Sleep interference was evaluated in four studies [32,42,43,45]. On the other hand, NP symptoms or sensory testing was reported only in one study [38], whereas the data were unclear in another two studies [42,43].

### 4.2. Quality of Evidence


In this review, before obtaining any results, we initially intended to perform a systematic review and meta-analysis. At first, we doubted if we should include in our quantitative analysis the studies already previously reviewed. As the NNT has changed, stabilizing after 2010 [10], we thought that a new separate independent and analytical review would be wiser. However, after the screening and selection, we found that a meta-analysis would not be possible. Thus, the quality of evidence has not increased after all these years. Nevertheless, some good-quality studies have demonstrated the superior efficacy of two-drug combinations against a placebo and against monotherapy. 

Ten studies had very small treatment groups [31,32,33,34,35,37,38,39,40,41]. The impact of small numbers on the effect cannot really be calculated, and it can overestimate treatment effects [48]. Half of the studies did not report the primary outcome (i.e., ≥50% or ≥30% pain reductions from baseline), more than half did not report a comparison with a period of experimental treatment versus the comparator for 12 weeks or longer, and the one study [45] with a comparison period of 12 weeks or longer did not report the primary outcome. It is noteworthy that a recommendation was already made in 2012: a sustained therapeutic effect in chronic pain should be demonstrated in pivotal efficacy trials with a treatment period of at least 12 weeks [47].

As we did not find several available studies with good-quality evidence for any one specific combination, we could not conduct any quantitative analyses even if we added the previous systematic review [27]. Therefore, we cannot make any recommendations on any specific drug combination for neuropathic pain over another.

### 4.3. Data from the Other Unselected Studies and Articles Assessed for Eligibility


As it was not possible to perform a systematic review and quantitative analysis from the selected studies, we took one step back and looked at evidence in the other 12 studies and other relevant open-label or observational studies published within this period, which were assessed for eligibility but not included for the qualitative synthesis. Even though the conclusions from those studies are not enough to make strong recommendations, they may be useful in guiding further studies. 

#### 4.3.1. Cannabinoids in Combination

One of the results of this review is that adding THC/CBD to a pre-existing treatment for NP did not show any benefits for these patients [43]. On the other hand, a recent review on only nabiximol (THC/CBD) for NP found that it was superior to the placebo but with a small effect size [49]. This small effect size alone may be the reason why it is not useful in combination. Moreover, there may be a difference between whether the CT is used with THC alone. Nabilone, a synthetic THC analogue, added to gabapentin (GBP) could be beneficial [39], but again, the results were produced from a very small number of participants, and the study was performed only for nine weeks. Findings indicating that GBP synergistically enhances THC have also been reported [50]. Thus, THC, but not a combination of THC/CBD, may represent a potential adjuvant for NP medications.

#### 4.3.2. Topical Treatments in Combination

Evidence for other topical treatments in CT is also controversial. Apart from the RCTs selected in this review, where CP8 was demonstrated to reduce NP as an add-on therapy [44], we also found three other non-selected studies on a lidocaine 5% plaster [51,52,53], one retrospective analysis on transdermal buprenorphine [54], and a very recent study protocol of a study combining clonidine and pentoxifylline [55]. One of the lidocaine 5% plaster studies [51] and the transdermal buprenorphine study not only were retrospective but also had a low number of participants, with different pain conditions and several concomitant therapies. Therefore, they were not suited for drawing any conclusions. Another retrospective study using a lidocaine 5% plaster as an off-label add-on therapy for different localized NP syndromes and conducted with 130 patients found that only 79 were still on the plaster after 3 months (44 after a year) [52]. Nevertheless, out of the 130, 66 reported >30% pain relief, from which 39 reported >50%. Despite being retrospective, this study suggests that lidocaine 5% plaster as an add-on therapy could have the same effect as CP8. Furthermore, an RCT for lidocaine 5% plaster against the placebo did pinpoint some findings about its use [53]. Randomization was stratified by concomitant treatment status, and no significant differences were found among the study groups, even though the treatment arm experienced better pain relief. A subgroup analysis showed that the add-on therapy group behaved almost the same as the placebo group. Hence, the results from the available literature on lidocaine 5% plaster are heterogeneous and inconsistent and should be clarified in a proper RCT for CT.

#### 4.3.3. Gabapentinoid and Opioid Combinations

The findings on the association of gabapentinoids with opioids are inconclusive. One RCT with a small number of participants and carried out for only 14 days showed that adding PGB to morphine in an N-CP was useful in reducing morphine dosage [33]. This morphine dose reduction was also suggested in a retrospective analysis [56]. However, the efficacy was the same between the CT and using morphine alone. An open-label study with morphine and PGB against both drugs in monotherapy under different NP conditions showed that CT was similar to morphine and superior to PGB in monotherapy [57], although this study, in addition to being open-labeled, had a very high drop-out rate in both monotherapy arms. A similar finding was reported for an eight-week non-inferiority RCT where tapentadol alone showed no difference when compared with a combination of tapentadol plus PGB [36]. No comparisons were made for the PGB alone or with the placebo. Unfortunately, even though this RCT showed a decrease in mean changes in pain intensity in both arms, the primary outcome for selection in this review was not shown, and we could not deduce it from the figures or tables. On the contrary, two open-label observational studies that added PGB to the pre-existing treatment [58,59] and another that added oxycodone/naloxone to patients already taking gabapentinoids [60] reported a decrease in pain. However, neither of them were actual RCTs (i.e., no placebo and no randomization). As a result, we can say that little to no difference is found in efficacy when combining gabapentinoids with opioids, whereas it may be a useful leverage for opioid dose reduction.

#### 4.3.4. Antidepressant and Opioid Combinations


The literature on antidepressants and opioids is limited too, but the results are more consistent towards a benefit for CT. In fact, combining antidepressants, be they tricyclic or otherwise, with opioids is a more frequent combination than combining antiepileptics with opioids [61]. Whilst DXT and methadone reduce cancer-related pain when compared with each drug alone (monotherapy) [62], adding DXT to an opioid–PGB therapy might have clinical benefits in alleviating refractory N-CP [32], and a superior efficacy of a nortriptyline–morphine combination has also been reported over each of these drugs in monotherapy [37]. Another RCT with a DXT–methadone combination could not be completed due to recruitment and retention issues [41]. Even though little evidence exists, the quality seems better for CT with antidepressants plus opioids than for gabapentinoids plus opioids.

#### 4.3.5. Gabapentinoids and Antidepressants in Combination


Another very frequently found combination is the one between gabapentinoids and antidepressants [61,63]. In fact, combinations of PGB/GBP and DXT/TCAs have been previously recommended for consideration as an alternative to increasing dosages in monotherapy for patients unresponsive to monotherapy with moderate dosages [8]. The American Academy of Neurology (AAN) guidelines have recommended adding venlafaxine to GBP in patients with inadequate pain relief on GBP monotherapy [64]. Recent evidence is contradictory, and recommendations may need to be reconsidered. An RCT selected in this review demonstrated the superiority for CT with PGB and imipramine. [38]. This RCT, though, had a low number of patients per arm, and the test lasted only for 5 weeks. The period was too short to show a persistence of the effect. Even so, PGB and imipramine in moderate doses was significantly superior to either drug in moderate-dose monotherapy. In another cohort study, PGB superadded to a pre-existing amitriptyline regimen helped to reduce pain [65]. However, though the authors claimed that the study was a randomized placebo-controlled study, the blinding and randomization procedures were not described properly. In addition, other recent contradictory results have been presented. In a post hoc analysis of another previous non-inferiority trial for DXT against PGB, patients treated with DXT plus GBP showed greater pain reduction than PGB monotherapy but not to DXT monotherapy, which was even more effective in patients who previously did not take any type of antidepressant [66]. Additionally, one of the other selected RCTs, the COMBO-DN study, did not find any statistically significant difference between the combination of DXT with PGB and high-dose monotherapy of either of them [42]. This RCT had some biases that made the results difficult to interpret. The result only lasted 8 weeks within the comparison period, no comparison for CT against low doses monotherapy was made, and the drop-out rate was high for several reasons: 109 out of 804 (13.5%) of the initial participants were drop-outs due to adverse effects, 10 dropped out due to a lack of efficacy, 42 dropped out due to patient decision, 64 dropped out for other reasons (noteworthily, 12 of them were withdrawn despite presenting a “satisfactory response” just before completion of the trial, and this issue, far from being odd, also appears in another selected study [44]), whereas only 290 completed the study out of 804 initially randomized. Additionally, in another cohort study, a combination of anticonvulsants and antidepressants was not associated with improved pain control at 6 months compared with individual therapy [67]. After considering these heterogeneous results, we are not sure recent evidence is strong enough to support recommendations on combining antidepressants with gabapentinoids. If a need for such combinations exists, the evidence shows it may be better to combine gabapentinoids with tricyclics.

Nevertheless, whether PGB should be added to the treatment of refractory uncontrolled pain, with an already broad treatment profile, remains to be answered. This was reported, with a relevant improvement in pain and treatment satisfaction, in two large observational studies [58,68], and neither TCAs nor opioids were found to be predictive factors for adverse events associated with PGB [69]. However, a re-analysis with pooled data from several RCTs showed that the therapeutic response to PGB was unaffected by concurrent NP medications and that the appearance of adverse events was unaffected too [70].

#### 4.3.6. Other Combinations


Finally, some interesting studies on other CTs have been performed, such as the combination of limaprost (prostaglandin E1 analog) with PGB, which did not provide additional relief in symptoms when compared with monotherapy with each of these drugs [35]. A combination of methadone and the N-methyl-D-aspartate (NMDA) antagonist ketamine was not better compared with methadone or ketamine alone [34], although the number of participants was low (14 on each arm). Either way, in a recent RCT, both ketamine alone and in combination with magnesium were found to not provide pain relief [31] despite a short 5-week study period. DXT and PGB, again in monotherapy, were compared, also recently, against a combination of either one with epalrestat (an aldose reductase inhibitor approved in some countries for the improvement of subjective neuropathy symptoms associated with diabetic peripheral neuropathy) [30]. That study demonstrated that PGB and epalrestat therapy had better effects on NP reduction than DXT and epalrestat within 3- and 6-month periods, but we could not figure out whether a significant difference against monotherapy was found. We also could not find information on the number of responders or drop-outs. The other RCT included for qualitative analysis but not for complete analysis due to not meeting the requirements compared two doses of a combination of dextromethorphan and quinidine [45]. A comparison against the placebo but not against monotherapy was made. Nevertheless, these drugs in combination are not among those recommended by clinical guidelines. Therefore, recommendations in this regard must be made with caution.

### 4.4. Implications for Clinical Practice


The burden of NP seems to be related to the complexity of neuropathic symptoms, poor outcomes, and difficult treatment decisions. Importantly, quality of life is impaired in patients with NP owing to increased drug prescriptions and visits to healthcare providers [71]. Published guidelines up until now recommend starting treatment with monotherapy [7,8,25,26]. If the first treatment is ineffective, the recommendation most frequently given is to switch drugs for another first-line treatment. However, some controversies exist about what to do in case of poor efficacy. After reaching the maximum tolerated dose, in clinical practice, for the management of NP, a second or even a third drug in combination regimens are frequently added [5,6,19].

Little evidence regarding CT exists. Despite the different treatment options available for NP, many patients do not experience clinically significant pain relief. In addition, they often experience adverse effects that make them unable to tolerate treatment [72]. Thus, clinicians often resort to concurrent administration of more than one pharmacological agent [58,73]. Combinations of analgesics used simultaneously in acute pain have been demonstrated to provide additive pain relief [74,75], and combination analgesics are among the most effective drugs in acute pain [76]. Given the evidence that a considerable number of patients with NP receive two or more drugs [61,63], we were only able to identify 16 recent relevant citations for this review and only 6 high-quality NP RCTs that evaluated the strategy of CT. Even more surprisingly, almost 10 years after the last review was published [27], these problems have not been addressed, and clinicians still need to rely on low-quality evidence and empirical knowledge when it comes to prescribing CT for NP. 

Nevertheless, with the current evidence, we suggest a flow diagram for those in need of starting CT. This proposal is based on the results of this review and is only intended to serve as a guide. Our aim with this review was not to make any recommendations. The flowchart begins with a patient who is already on antidepressants or on opioids as concurrent medication (Figure 2). The option for those who are already on gabapentinoids or on duloxetine is not shown. Evidence in this regard is inconclusive and controversial.

### 4.5. Implications for Research


In order to properly identify specific CT that provides superior efficacy and/or safety, we recommend that future NP studies of two-drug combinations include comparisons with placebo and both single-agent components. When designing the study protocol, before calculating the sample size, researchers should consider that pain RCTs have a higher placebo response [15] and that claims about an increase in drop-out numbers have been made [38,41,42]. Moreover, a crossover trial takes longer than a parallel one, increasing the chance of more dropouts. 

In addition, we encourage NP guidelines to include recommendations of which NP CT to study, so that better evidence can be reached, and meta-analyses can be conducted afterwards. Reports of widespread clinical NP CT benefits provide an impetus for additional future investigations. Regarding this matter, a demonstration of CT benefits by several studies in animals could also provide a rationale for studies in this and other directions [50,77,78,79,80,81,82,83,84,85].

For instance, in non-clinical studies, the potentiation of morphine by GBP has been validated in a chronic constriction injury model of NP [77,78]. Likewise, the combination of GBP and tramadol in a partial sciatic nerve ligation model [79], peripheral neuropathy induced by paclitaxel [80], and diabetic neuropathy [81] has been validated. Furthermore, no significant drug-to-drug interaction between PGB and tramadol has been studied in healthy volunteers [86], and surprisingly, a recent proposal to make a compound tablet with tramadol and GBP was put forward [87], even though these combinations have not yet been validated in proper RCTs.

Similarly, other combinations have been also tested in animal models. THC (with no CBD) and GBP reduced mechanical and cold allodynia in a chronic constriction injury model but without diminishing the THC-related side effects [50]. Another pan-cannabinoid receptor agonist, when administered together with morphine, reduced allodynia in a synergistic manner but had only an additive effect on motor incoordination [82]. The same agonist had supra-additive effects on cold allodynia in a post-operative model combined with a selective noradrenaline reuptake inhibitor [83].

Finally, researchers could also try different combinations. CT with two or more classes of antiepileptics is common in clinical practice for epileptic disorders. This combination has not been fully explored in NP. Regarding this matter, some results have been found in a nerve ligation model, where carbamazepine and PGB synergistically ameliorated NP at higher doses [84]. In addition, NMDA receptor antagonists, together with GBP, have also provided synergistic effects in the alleviation of NP in a SCI model, while reducing side effects [85].

Moreover, research on phenotypes responding to treatment may provide further suggestions about CT for NP. Even among individuals with seemingly singular neuropathic conditions (e.g., PHN), substantial diversity exists with respect to various clinical manifestations, sensory examination features, and presumably underlying pain mechanisms [21,22]. For instance, recently, Benavides et al. found a functional polymorphism that could predict pharmacologic response to a combination of nortriptyline and morphine in NP patients [88].

### 4.6. Potential Biases and Limitations


We tried to scope the results of this review in the most objective way possible. However, we had some difficulties finding data. First, some trials were found after the third or even the fourth database search. Therefore, though unlikely, the possibility of missing RCTs still exists. As we could not afford the fee to search EMBASE, we may have missed information. Additionally, as the search was already very large, we did not include other websites (e.g., controlled-trials.com, and clinicalstudyresults.org). Thus, we may have missed some RCTs. This can lead to a publication bias. However, we considered that, as the search was conducted on four major databases with over 2000 different citations and over 1500 duplicated results, the probability of missing a trial was very low and the amount of work and duplicates would increase even further. Another difficulty we found was looking for proper data within the publications. Some data were very accessible, but other data needed to be inferred via tables, figures, or even the discussion. Thus, even after a colleague (see the Acknowledgments) reviewed our work, we may have made mistakes with the discernment of extracted data.

Another serious limitation was complying with all items for a systematic review. We did not register the protocol for this review in a review registry (e.g., PROSPERO). This is a critical flaw according to AMSTAR-2 [89]. Even so, we have tried to reflect, in the material and methods section, the entire search protocol as it was carried out. For conducting a proper systematic review, we missed item 2 (protocol registration), whereas items 4 (literature search) and 7 (justification for excluding individual studies) were partially fulfilled, but it only remains an issue for reviewing those studies conducted only after 2012.

We also could have performed a thorough meta-analysis including all the RCTs published prior to 2012 [27]. By doing this, we may have achieved a systematic review and a quantitative meta-analysis. Chaparro et al. performed one with only two RCTs. However, given the changes in trial methodology and requirements by EMA and FDA and that the NNT has increased (accompanied by a decrease in effect size), which stabilized after 2010 [10] we thought that not including those studies would be wise. Even though this may lead to a discussion about publication bias, mixing those trials could generate more bias and confusion than benefits. Maybe we should have been more thorough and included studies published after 2010. However, trying to conduct a complete meta-analysis based on individual data, not by sizing up several RCTs but by shelling individual data from every RCT and only then conducting the complete meta-analysis, would be wiser. This work is very time-consuming simply regarding retrieving individual data from old records, and we could not afford to do so. We recommend that other researchers follow this path for the purpose of obtaining better evidence on CT for NP.

Finally, we decided to keep the primary outcome strictly as the proportion of participants reporting ≥50% pain reductions from baseline (or ≥30% when 50% was not reported) and did not add other pain evaluations such as moderate or greater pain relief, or moderate or greater overall improvement, for the purpose of obtaining stronger evidence. In fact, the latest review used these other options as primary outcomes [27]. If we had added these criteria in the primary outcome, we could have gathered more studies. However, we believe that a study with measures of pain reduction as the primary outcome but not reporting the number of participants with that pain reduction in their results to be awkward. Hence, not reporting these numbers would give these awkward studies the same status as other studies that did report such an endpoint. If regulatory agencies have raised their standards for conducting RCTs from ≥30% to ≥50% pain reductions from baseline, we inferred that a systematic review should raise the primary endpoint too. Consequently, we made the primary objective simple and highly selective. As said before, the risk of doing so is leaving some good studies out. However, reporting the number of participants with a pain reduction is an important issue in pain trials. Reporting the number of participants with a reduction in other scores such as overall improvement or pain relief is also useful, but reporting the most important outcome should not be compromised. 

### 4.7. Agreements or Disagreements with Other Studies or Reviews


We completely agree with Eisenberg and Suzan’s review. Even though several new trials have used various drug combinations for NP, the results are still inconsistent due to methodological problems [24]. We partially agree with the review by Finnerup et al. [8]. In recent years, no trials have been conducted on GBP CT. A combination of PGB with TCAs may be one option, whereas its combination with DXT (or other selective serotonin noradrenaline reuptake inhibitors) has yet to be elucidated.

In a review on topical treatments for localized neuropathic pain conducted by Casale et al., the evidence to support systematic use as treatment options was still insufficient [90]. Now, after reviewing the selected and other recent non-selected studies, we think that CP8 may be used systematically as an add-on therapy. 

We do not agree completely with the conclusions of Guan et al. regarding anticonvulsants or antidepressants, in which they claim that CT reduces NP [91]. Even though their systematic review and meta-analysis was conducted only for NP in cancer patients, only three of the eight selected studies used drug combinations as experimental compounds. We found that the evidence for this topic remains controversial.

Again, more than nine years after the last review, we continue to agree with the conclusions of that last systematic review [27]. For this period, the total number of citations may have increased, but the number of high-quality NP RCTs that evaluated the strategy of CT has not. Again, in our review, only one eligible study evaluated a combination of the two most widely used classes of neuropathic pain drugs, i.e., antidepressants and anticonvulsants. Additionally, once again, the paucity of recent available studies for each drug class combination studied from the last review until now precludes any well-founded conclusions about most combinations. The search strategy for this review was not designed to capture all studies available to date but only those published after 2012; therefore, another review that includes all studies published to date may produce different conclusions. However, as mentioned before, we designed this review according to changes that may have influenced RCTs. Combining recent and older RCTs may also generate confusion.

## 5. Conclusions

Neuropathic pain treatment continues to be an unmet medical need, as patients keep reporting inadequate pain relief. Clinicians continue to have problems dealing with how to face pharmacological strategy when first-line treatment fails. CT has been a practice adopted for many years for which the evidence is not solid. Efforts have been made to achieve better-quality evidence, but the quality has not improved over the years. Guidelines for neuropathic pain should attempt to make recommendations about CT research, prioritizing which combinations to analyze over others, so that the search for better evidence can take steps forward.

## Figures and Tables

**Figure 1 jcm-10-03533-f001:**
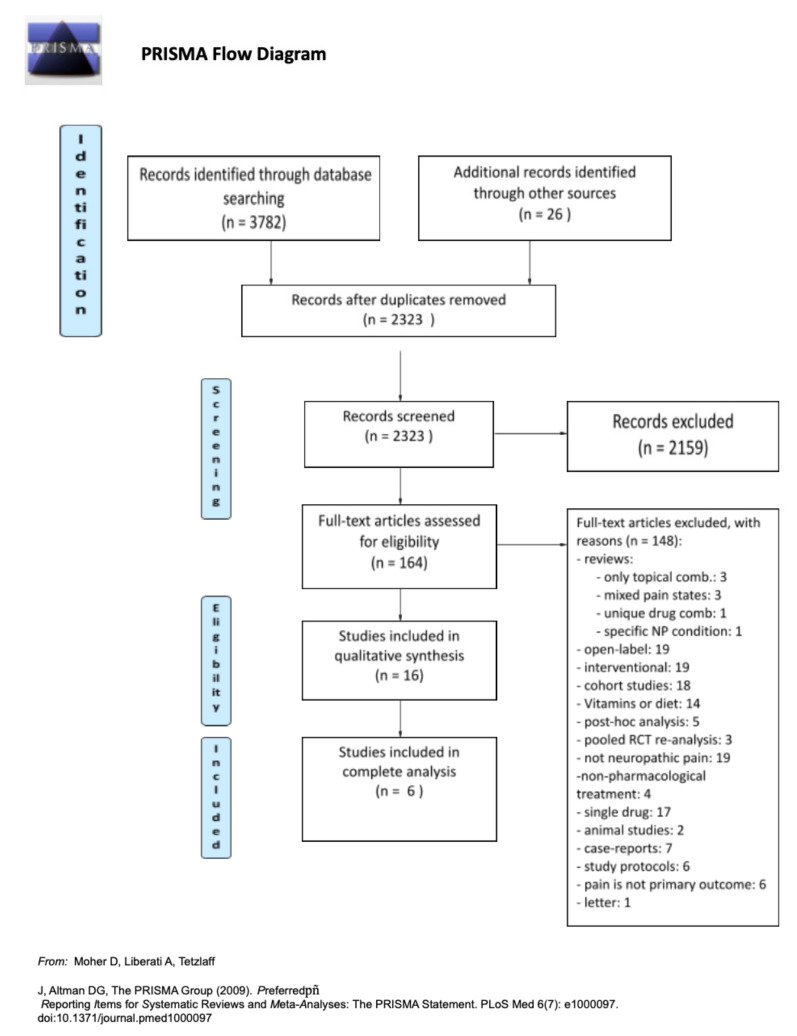
PRISMA flow diagram.

**Figure 2 jcm-10-03533-f002:**
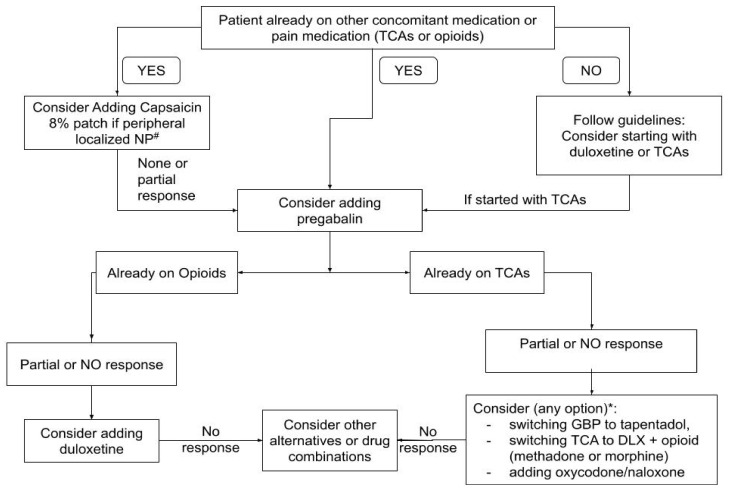
A guide presented as a flowchart for a strategy for combination therapy for neuropathic pain. This flowchart is a strategic proposal for prescribing combination therapy for neuropathic pain that begins with a patient already on opioids or tricyclic antidepressants. The option for those who are already on gabapentinoids or on duloxetine is not shown. Evidence in this regard is inconclusive and controversial. This flowchart is a proposal drawn from the recent evidence presented in this review. Clinicians should consider that this guide may (and should) change as new evidence is brought to light. ^#^ = While in this review we did not find published evidence on lidocaine patches, these should be considered where appropriate by previously published evidence. * = Evidence on opioid combination therapy is controversial and inconclusive. From recent evidence, these seem to be the best available options. TCAs = tricyclic antidepressants, NP = neuropathic Pain, DLX = duloxetine.

**Table 1 jcm-10-03533-t001:** Data from the selected studies.

Name	Pain Condition	RCT	Treatment Duration (Weeks)	Combination	Target Ceiling Dose or MTD per Day	Route	Sample Size RD (CS)	Control	Target Ceiling Dose or MTD per Day	Route	Sample Size RD (CS)
Langford 2013 [43]	Central neuropathic pain in patients with multiple sclerosis	DB; PARALLEL	14	**THC/CBD + concomitant analgesic medication**	32.4/30 mg	oromucosal (spay) + oral	167 (41)	Placebo		oromucosal (spray) + oral	172 (156)
Shaibani 2012 [45]	Diabetic neuropathic pain	DB; PARALLEL	13	**DMQ**	90/60 mg 60/60 mg	oral oral	131 (79) 125 (74)	Placebo		oral	123 (89)
Irving 2012 [44]	Postherpetic neuralgia	DB; PARALLEL	12	**Capsaicin + concomitant neuropathic medication**	640 µg/cm^2^	topical (skin) + oral	597 (544)	Placebo		topical (skin) + oral	530 (480)
Tesfaye 2013 [42]	Diabetic neuropathic pain in patients who are non-responders to duloxetine or pregabalin	DB; PARALLEL	8	**Duloxetine + Pregabalin**	60 + 300 mg	oral	170 (141)	Duloxetine Pregabalin	120 mg 600 mg	oral oral	74 (?) 99 (?)
Holbech 2015 [38]	Painful polyneuropathy	DB; CROSSOVER	5	**Imipramine + Pregabalin**	75 + 300 mg	oral	18 (15)–16 (15) –15 (12)–16 (14)	Placebo Imipramine Pregabalin	75 mg 300 mg;	oral oral oral	19 (18)–16 (15)–15 (13) 12 (11) 18 (17)–17 (14)–14 (14)–12 (12) 18 (15)–16 (14)–14 (14)–13 (13)
Matsuoka 2019 [32]	Neuropathic pain in cancer patients who are non-responders to opioid–pregabalin	DB; PARALLEL	1,5 (10 days)	**Duloxetine + Opioid–Pregabalin**	40 mg + ?—300 mg	oral	35 (34)	Placebo + Opioid- Pregabalin	?- 300 mg	oral	35 (33)

RCT: randomized controlled trial; MTD: maximum tolerated dose; RD (CS): randomized (completed study); DB: double-blind; THC/CBD: tetrahydrocannabinol/cannabidiol; DMQ: dextromethorphan + quinidine; ?: data not available.

**Table 2 jcm-10-03533-t002:** Data from the non-selected studies.

Name	Pain Condition	RCT	Treatment Duration (Weeks)	Combination	Target Ceiling Dose or MTD per Day	Route	Sample Size RD (CS)	Control	Target Ceiling Dose or MTD per Day	Route	Sample Size RD (CS)
Singh 2021 [30]	Diabetic neuropathic pain	DB; PARALLEL	24 (6 months)	**Epalrestat + pregabalin Epalrestat + duloxetine**	100 + 150 mg 100 + 60 mg	oral oral	50? (?) 50? (?)	Pregabalin Duloxetine	150 mg 60 mg	Oral oral	50? (?) 50? (?)
Rigo 2017 [34]	Neuropathic pain in patients whose responses to neuropathic medication are poor	DB; PARALLEL	13 (3 months)	**Methadone + ketamine**	9 + 90 mg	oral	14 (13)	Methadone Ketamine	9 mg 90 mg	oral oral	14 (13) 14 (11)
Turcotte 2015 [39]	Central neuropathic pain in patients with multiple sclerosis and treated with gabapentin	DB; PARALLEL	9	**Nabilone + gabapentin**	2 + 1800 mg	oral	8 (7)	Placebo + gabapentin	1800 mg	oral oral	7 (7)
Kim 2016 [35]	Lumbar spinal stenosis	DB; DD; PARALLEL	8	**Limaprost + pregabalin**	15 µg + 225 mg	oral	61 (43)	Limaprost Pregabalin	15 µg 225 mg	oral oral	61 (40) 60 (43)
Baron 2014 [36]	Low back pain (with a neuropathic component) in patients treated with tapentadol PR	DB; PARALLEL	8	**Tapentadol PR + pregabalin**	300 + 300 mg	oral	159 (133)	Tapentadol PR	500 mg	oral	154 (126)
Gilron 2015 [37]	Neuropathic pain	DB; CROSSOVER	6 (period)	**Nortriptyline +** **Morphine**	100 + 100 mg	oral	15 (13) –11 (9)–18 (15)	Nortriptyline Morphine	100 mg 100 mg	oral oral	13 (13)–16 (14)–16 (16) 17 (14) –14 (10)–16 (14)
Pickering 2020 [31]	Neuropathic pain (long-standing refractory)	DB; CROSSOVER	5 (period)	**Ketamine + magnesium**	0.5 mg/kg + 3g	i.v.	20 (20)	Placebo Ketamine	0.5 mg/kg	i.v. i.v.	20 (20) 20 (20)
Harrison 2013 [41]	HIV-associated polyneuropathy	DB; CROSSOVER	4 (period)	**Duloxetine + methadone**	60 + 30 mg	oral	4 (3) -3 (3)?–3 (3) –4 (3)?	Placebo Duloxetine Methadone	60 mg 30 mg	oral oral oral	4 (4)? –4 (3)?–2 (2)? –4 (3) 4 (4) -3 (3)?–3 (2) –4 (4)? 4 (4)–4 (2)–2 (2)? –3 (3)?
Garassino 2013 [40]	Neuropathic pain in cancer patients	PARALLEL	2	**Pregabalin ↑ + oxycodone fix** **Pregabalin fix + oxycodone ↑**	300 + 20 mg 50 + 20 mg?	oral oral	38 (32) 37 (35)				
Dou 2017 [33]	Neuropathic pain in cancer patients treated with morphine	DB; CROSSOVER	2 (period)	**Pregabalin + morphine PR**	300 + ≥180 mg	oral	20 (?)–20 (?)	Placebo + morphine PR	+ ≥ 180 mg	oral	20 (?)–20 (?)

RCT: randomized controlled trial; MTD: maximum tolerated dose; RD (CS): randomized (completed study); DB: double-blind; DD: double dummy; DMQ: dextromethorphan/quinidine; PR: prolonged or sustained release; ?: data not available.

**Table 3 jcm-10-03533-t003:** Risk of bias.

Study	Bias Arising from the Randomization Process	Bias Due to Deviations from Intended Interventions	Bias Due to Missing Outcome Data	Bias in the Measurement of the Outcome	Bias in the Selection of the Reported Result	Other Potential Sources of Bias
Langford 2013 [43]	+	+	+	+	?	?
Irving 2012 [44]	+	+	+	?	?	?
Shaibani 2012 [45]	+	+	+	+	+	+
Tesfaye 2013 [42]	+	+	+	+	?	-
Holbech 2015 [38]	+	+	+	+	?	-
Matsuoka 2019 [32]	+	+	+	+	?	-
Singh 2021 [30]	?	-	-	+	?	-
Rigo 2017 [34]	+	+	+	+	?	.
Kim 2016 [35]	+	+	+	+	?	-
Baron 2014 [36]	?	+	+	+	?	-
Gilron 2015 [37]	+	+	+	+	?	-
Pickering 2020 [31]	+	+	+	+	?	-
Turcotte 2015 [39]	+	-	+	+	+	-
Harrison 2013 [41]	+	-	+	+	?	-
Dou 2017 [33]	+	+	+	+	?	-
Garassino 2013 [40]	?	-	+	?	?	**-**

This table reproduces our judgements about each risk of bias item for each study. + with a green background stands for a low risk of bias. ? with a yellow background stands for unclear risk of bias. - with a red background stands for a high risk of bias.

## Data Availability

Not applicable.

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
