# Peer review of "Combination Therapy for Neuropathic Pain: A Review of Recent Evidence"

_jcm, 2021, doi:10.3390/jcm10163533_

Round 1

Reviewer 1 Report

The authors have submitted a manuscript entitled “Combination Therapy For Neuropathic Pain. A Systematic Review” which, as the authors explain in the Introduction, appears to follow a 2012 Cochrane review (Chaparro et al, 2012) with a very similar title.

The submitted review does not include any studies prior to 2012 (i.e. studies that were included in Chaparro et al, 2012). Therefore, it would seem that the authors intend this manuscript to be an update to a previous review conducted by a different group of authors. However, the authors do not describe this either as 1) a review *update* or, 2) a new independent review unrelated to the Cochrane review by Chaparro et al. This is a critical issue that needs to be clarified. 

The authors do not state whether or not their review methods are the same as those of the Chaparro et al 2012 review. This is important because if different methods are used, this may be one reason for different findings. Also, it is not clear whether previous evidence from the 2012 was considered for possible updates to any meta-analyses that were previously conducted. 

This review uses different outcomes than the 2012 review. Also, by failing to include a measure of global improvement or global pain relief – as has been done previously – this review needlessly excludes several potentially informative studies that could contribute data to one or more meta-analyses of trial results.

 There is no mention of whether or not more than one reviewer searched, screened, evaluated and conducted data extraction on the studies in this review.

 Multiple studies in Table 2 actually do include data on global improvement and/or global pain relief that are relevant to the primary outcome and appear to be incorrectly excluded from the review.

 It does not appear that a protocol for this review had been previously defined or registered in a review registry (e.g. PROSPERO). This is a critical flaw according to AMSTAR-2.

 The authors do not provide a complete literature search strategy and what they do describe is not transparently and explicitly shown in such a way that anyone could replicate the search results.

 Beyond the ‘non-selected studies’ shown in Table 2, no other excluded studies are reported, for example, was Wang et al., 2017 considered? (Wang et al.,. Morphine and pregabalin in the treatment of neuropathic pain. Exp Ther Med 2017;13:1393). If it was, why was it excluded? There are multiple other studies like this that should be reported to indicate that they were considered and if so, why they were excluded.

 The Risk of Bias table (Table 3) does not have a legend to explain the meaning of the different colors (red, green, yellow). There are several studies that have ‘green’ and/or ‘yellow’ for most of the criteria, however, they have ‘red’ for the Overall Bias. These does not seem correct and needs to be corrected and clarified. Also, it is unclear why risk of bias has been scored for the 10 studies that the authors decided were ‘non-selected’.

 There is a brief section (3.3) that superficially mentions the results for the 6 studies the authors decided were included. However, no quantitative data are provided. This is the most critical element of a systematic review and it seems to have been omitted from the manuscript.

 Discussion and Abstract:

Given the several important flaws in the methodology and reporting in this review, the interpretation and discussion of results presented here is not appropriate since several studies were incorrectly excluded.

Reviewer 2 Report

Thanks to the authors for reviewing this interesting review related to combination therapy for neuropathic pain.
The paper appears well structured; in particular, I appreciated the following points more:
- completeness of the systematic review of the literature both in terms of bibliography and explanation of the analysis
- identification of a clear and clinically significant outcome.
- synthesis of very complex results in an understandable way.
I, therefore, believe that the work in its entirety is of absolute value.
Allow me to highlight just a few points:
- page 1 introduction: there is a typo with twice written "it negatively affects quality of life ..."
- page 2. it is not clear to me what is meant in line 6: "This results in a decrease ....... around 2010"
- p. 4 2.7.8 The reason why a meta-analysis was not carried out is well expressed in the article, but in this paragraph, it seems unclear. I would evaluate whether to remove it, leave the topic in the discussion, or be more detailed here and less in the discussion.
- par 3.3 effects of interventions: In this case, it seems a concise summary of what was then taken up in a much more complete way. From my point of view, it would lighten the reading delete it, but it is not wrong in itself.
- par 4.3.3 gabapentinoids and opioids combination: I would eliminate the phrase "perhaps we could say that they might speak against the CT" because it is no longer justified. I would leave only "inconclusive."
- in the proposed flow chart (clearly explaining that this is not a guideline), it seems little justified to speak only of the capsaicin patch (correctly resuming the work of Casale et al.) by not including the lidocaine patch. I believe that if the topical part is also treated, so much to include it in the flow-chart, it cannot be lacking in treatments unless they have been evaluated as ineffective.
I still take this opportunity to thank the authors for the excellent work that I had the honor to read and evaluate in preview.

Round 2

Reviewer 1 Report

The authors have submitted a revision of their original manuscript entitled “Combination Therapy For Neuropathic Pain.

There remain several serious problems (discussed below) with this manuscript – if it is to be considered a true “Systematic Review”. Two options to deal with this could include:

  1. To conduct a formal systematic review that follows AMSTAR-2 guidelines (Shea et al., BMJ 2017;358:j4008) starting with registering a new review protocol e.g. in a registry like PROSPERO (https://www.crd.york.ac.uk/PROSPERO/), and conducting a FULL review of ALL published trials (not just since 2012);

OR

  1. To revise the title of this manuscript to “Combination Therapy for Neuropathic Pain: A Brief Review of Recent Studies” – OR – “Combination Therapy for Neuropathic Pain: A Description of Recent Evidence”.

Comments:

This review considers only studies published since 2012 with no consideration of previous evidence in this area. Unfortunately, by failing to consider ALL available studies, i.e. including those published prior to 2012, this review does not recognize multiple groups of similar studies (e.g. opioid+gabapentinoid, opioid+antidepressant, gabapentinoid+antidepressant) that could be combined for meta-analysis. Therefore, this manuscript fails to achieve one of the most important goals of a systematic review, i.e. considering ALL available evidence and combining this evidence to provide a more robust estimation of treatment efficacy and safety.

Unfortunately, if this manuscript is to be considered as a true Systematic Review it has several very important serious flaws making it a Critically Low Quality review according to AMSTAR-2 (Shea et al., BMJ 2017;358:j4008). These critical flaws include:

  1. a) the authors did not register a protocol for the methods used in this review (the authors acknowledge that this review did not exactly follow the same protocol as the 2012 Chaparro review and they do not justify/explain changes made to that previous protocol);
  2. b) despite some general information provided in section 2.6, the authors do not provide a complete search strategy for the review;
  3. c) the authors do not provide an adequate explanation for failing to conduct any meta-analyses despite some groups of similar studies that could undergo meta-analysis;
  4. d) the authors to not address the possibility of publication bias.
